# Biologically Relevant Murine Models of Chronic *Pseudomonas aeruginosa* Respiratory Infection

**DOI:** 10.3390/pathogens12081053

**Published:** 2023-08-17

**Authors:** Aoife M. Rodgers, Jaime Lindsay, Avril Monahan, Alice V. Dubois, Aduragbemi A. Faniyi, Barry J. Plant, Marcus A. Mall, Miquel B. Ekkelenkamp, Stuart Elborn, Rebecca J. Ingram

**Affiliations:** 1Wellcome-Wolfson Institute of Experimental Medicine, Queen’s University Belfast, Belfast BT7 1NN, UKs.elborn@qub.ac.uk (S.E.); 2Cork Centre for Cystic Fibrosis (3CF), Cork University Hospital, University College Cork, T12 E8YV Cork, Ireland; 3The HRB Funded Clinical Research Facility, University College Cork, T12 E8YV Cork, Ireland; 4Department of Pediatric Respiratory Medicine, Immunology and Critical Care Medicine, Charité—University of Medicine Berlin, 10117 Berlin, Germany; 5German Center for Lung Research (DZL), 10117 Berlin, Germany; 6Berlin Institute of Health at Charité—University of Medicine Berlin, 10117 Berlin, Germany; 7Department of Medical Microbiology, University Medical Center Utrecht, 3584 CX Utrecht, The Netherlands

**Keywords:** *Pseudomonas aeruginosa*, chronic infection, respiratory, murine model, in vivo, *S. aureus*

## Abstract

*Pseudomonas aeruginosa (P. aeruginosa)* is an opportunistic pathogen and the leading cause of infection in patients with cystic fibrosis (CF). The ability of *P. aeruginosa* to evade host responses and develop into chronic infection causes significant morbidity and mortality. Several mouse models have been developed to study chronic respiratory infections induced by *P. aeruginosa*, with the bead agar model being the most widely used. However, this model has several limitations, including the requirement for surgical procedures and high mortality rates. Herein, we describe novel and adapted biologically relevant models of chronic lung infection caused by *P. aeruginosa*. Three methods are described: a clinical isolate infection model, utilising isolates obtained from patients with CF; an incomplete antibiotic clearance model, leading to bacterial bounce-back; and the establishment of chronic infection; and an adapted water bottle chronic infection model. These models circumvent the requirement for a surgical procedure and, importantly, can be induced with clinical isolates of *P. aeruginosa* and in wild-type mice. We also demonstrate successful induction of chronic infection in the transgenic βENaC murine model of CF. We envisage that the models described will facilitate the investigations of host and microbial factors, and the efficacy of novel antimicrobials, during chronic *P. aeruginosa* respiratory infections.

## 1. Introduction

Cystic fibrosis (CF) is a life-shortening genetic disease caused by a mutation in a gene that encodes the cystic fibrosis transmembrane conductance regulator (CFTR) [1]. In CF patients, chronic infections caused by *P. aeruginosa* cause significant morbidity and mortality [1]. Initially, *P. aeruginosa* infections can be cleared by antimicrobial therapy. However, in most patients with CF, a transition to chronic infection by biofilm-forming mucoid strains of *P. aeruginosa* occurs, which persists, eventually resulting in respiratory failure [2]. Chronic infections in patients with CF are characterised by the development of *P. aeruginosa*-specific antibodies, chronic inflammation dominated by neutrophils, and increased release of serine proteases causing lung tissue damage, in addition to the damage caused by the bacteria [2,3].

Over the last several decades, several animal models have been developed in attempts to fully elucidate the bacterial pathogenesis during chronic infection caused by *P. aeruginosa* [4]. The administration of *P. aeruginosa* to the murine lung either causes acute infection and subsequent sepsis or is rapidly cleared from the lung [5]. Chronic infection has been achieved by intratracheal instillation using an immobilising agent. Cash et al. established a chronic model of respiratory infection with *P. aeruginosa* utilising agar beads administered intratracheally, which resulted in infection lasting up to 35 days [6]. The embedding of *P. aeruginosa* within agar beads facilitates bacterial retention, thus resembling bacterial biofilm formation, as evident in chronic infection. This approach is, however, technically challenging, necessitates the use of surgical procedures and injectable anesthetic agents, and has been reported to have mortality rates of up to 45% [7]. Hoffman et al. utilised a stable mucoid CF sputum isolate with hyperproduction of alginate to develop a chronic model of *P. aeruginosa* lung infection. The hyperproduction of alginate was induced by deletions in *mucA* and functional *N*-acylhomoserine lactone-based quorum sensing systems; when instilled in mice, they could cause infections that could last for up to 7 days [8]. In another approach, plastic tubes precoated with *P. aeruginosa* inserted into the bronchus were successfully used to establish chronic infection, which lasted up to 120 days [9]. More recently, Bayes et al. developed and adapted a murine agar bead model, using a clinical mucoid strain to induce features of transition from transitory to chronic airway infection [10]. However, it has been reported that bead size affects the induced response in this model, causing significant differences in inflammatory responses in the lung [11]. Furthermore, it has been reported that these models can result in chronic infection in the conducting airways due to blockage of the bronchi with beads.

To date, murine models of chronic *P. aeruginosa* infection without the use of artificial embedding have not been extensively reported. Novel animal models that recapitulate chronic infections caused by *P. aeruginosa* are urgently warranted to fully understand host pathogen-interactions and facilitate the development of new therapeutic targets. Here, we describe experimental conditions to establish biologically relevant murine models of chronic lung bacterial infection. Specifically, we report the establishment of chronic infection through the selection of clinical isolates, selection of murine strain, and optimisation of infection regime. Moreover, we report an incomplete antibiotic clearance model and adapted oral challenge water bottle [12]. Importantly, our models circumvent the requirement of artificial materials, overcoming challenges associated with currently reported models.

## 2. Materials and Methods

### 2.1. Mice

Wild-type mice were either purchased from Inotivco (Berkshire, UK) (A/J, BALB/c) or Charles River (Kent, UK) (CD-1, FVB/N, SJL, Biozzi, C57BL6), or then bred on-site in the Biological Services Unit in Queen’s University Belfast. Both male and female βENaC mice [13,14,15,16] between 8 and 18 weeks were used, and all groups were sex matched. Mice were housed in groups in individually ventilated cages with 12 h light and dark cycles and fed water and diet ad libitum. All animals were randomly assigned to treatment groups, using the GraphPad Prism tool (GraphPad Software, Boston, MA, USA). Mice were handled using Biosafety Level 2 practices.

### 2.2. Biofilm Assay 

Biofilm formation of bacterial strains was quantified using crystal violet staining. Briefly, 1 mL of overnight cultures were washed, resuspended in TSB supplemented with 1% glucose, and adjusted to an OD_600_ of 0.1. The culture suspensions were added to the wells of a 48-well tissue culture plate (Sarstedt, Nümbrecht, Germany) and incubated overnight without agitation at 37 °C. Subsequently, all media was removed from wells, then 150 µL of methanol was added and the plates were incubated for 20 min at room temperature. The methanol was then discarded; once the plate had air dried, 150 µL of 20% w/v crystal violet was added to each well for 5 min. Plates were washed thoroughly with water and left to dry completely. The remaining crystal violet was eluted with 150 µL of glacial acetic acid and measured at an OD_595_.

### 2.3. Intra-Nasal Infection of Mice

Mice were infected intranasally as previously described [17]. Mice were infected with clinical isolates of *P. aeruginosa* (provided by the Clinical Microbiology Laboratory in the Royal Victoria Hospital) or the commonly used laboratory strain PAO1 (20). The clinical isolates of Pseudomonas were passaged through CD1 mice three times prior to use in these experiments to enhance their adaptation to a murine host. The bacteria were grown to mid-log phase in nutrient broth and washed extensively using endotoxin-free sterile phosphate buffered saline (PBS). They were then resuspended in endotoxin-free sterile PBS and diluted to the appropriate optical density (OD_600_). The number of colony-forming units (CFU) was counted by plating serial dilutions of *P. aeruginosa* on nutrient agar (Thermotech, Nottingham, UK) plates. Mice were anesthetised with a mixture of Rompun-ketamine injected intraperitoneally. Once anesthetised, mice were held up-right and 20 µL of bacterial suspension in endotoxin-free PBS was dropped onto alternate nostrils and inhaled. Mice were held in an up-right position to allow the suspension to reach the lungs until their breathing rate returned to normal. Eye gel (Medicom Healthcare, Windsor, UK) was applied, and mice were gently placed into the cages with a Cocoon Bedding “pillow” (Ketchum, Lake Luzerne, NY, USA) propped beneath their chin to maintain their airway. Animals were monitored regularly to determine if they displayed any signs of illness including ruffled fur/hunched posture, significantly abnormal respiration, and mobility reduced to a few steps. If mice displayed any two of these symptoms or had weight loss greater than 20% of pre-procedure body weight, they were culled. At predetermined time points, mice were given a terminal dose of Rompun-ketamine anaesthetic and once fully unresponsive, exsanguination was completed by cardiac puncture; lung and nasal associated lung tissue (NALT) were collected. Tissue was homogenised by passing through a cell sieve, or mechanically processed using a Precellys Evolution (Bertin-technologies, Paris, France) with 1 cycle of 10 s at 4500 rpm. The homogenate was either serially diluted and plated onto cetrimide selective agar (Thermotech, Nottingham, UK) to quantify the bacterial burden or centrifuged and the cell pellet used to analyse the cell populations present by flow cytometry, and the supernatant was frozen for subsequent cytokine analysis.

### 2.4. Preparation of Pseudomonas Agar Beads and Oro-Tracheal Adminstration

Overnight culture of PAO1 was diluted to OD2 in 20 mL of Tryptone Soya Broth (TSB) (Thermotech, Nottingham, UK). This was incubated at 37 °C with shaking until it reached 10 OD (approximately 4 h); it was then centrifuged at 2700 g for 15 min at 4 °C. The supernatant was discarded and the pellet was resuspended in 1 ml of sterile PBS. Tryptone Soya Agar (TSA) was prepared by mixing TBS with 1.5% agar (Thermotech, Nottingham, UK); this was autoclaved and equilibrated to 50 °C. The bacteria were diluted 1 in 10 in TSA, this was then added immediately to 150 mL of preautoclaved heavy mineral oil at 50 °C in an Erlenmeyer flask, which was being continually stirred at room temp. After 5 min, the flask was transferred to 4 °C and stirred for a further 35 min. The beads-in-oil preparation was placed on ice for 30 mins, and then transferred to 50 mL falcon tubes. The oil was removed by washing 6 times by centrifuging at 2700× *g* for 15 min at 4 °C, discarding the supernatant and resuspending in sterile PBS. An aliquot of beads was aseptically homogenised and plated by serial dilution on nutrient agar (Thermotech, Nottingham, UK). The beads were stored overnight at 4 °C and then diluted in sterile PBS to 5 *×* 107 CFU/mL. Mice were anesthetized, and intubated oro-tracheally with an Insyte IV Catheter (Becton Dickenson, Plymouth, UK) and 50 uL of the bead preparation delivered into the lungs. The animals were culled 7 days after instillation and the bacterial burden within the lung determined by serial dilution of lung homogenate on selective cetrimide agar (Thermotech, Nottingham, UK).

### 2.5. Staphlococycuccus aureus Assessment

Staphylococcal strains were cultured at 37 °C in a Brain Heart Infusion (BHI) (Oxoid, Hampshire, UK). To evaluate protease activity, 5 μL of each strain adjusted to an OD_600_ of 1.0 was spotted onto BHI containing 1% skimmed milk. Plates were incubated at 37 °C for 48 h and positive protease activity was recorded based on zones of clearance in the media. The haemolytic activity of strains was evaluated on sheep blood (TCS Biosciences, Buckinham, UK) agar plates (Thermotech, Nottingham, UK) incubated at 37 °C for 48 h. Carotenoid pigment production was assessed for each strain by pelleting stationary phase cells by centrifugation. Production of the pigment was based on the colour of the pellet, where white represented negative, and orange represented positive. To assess antibiotic susceptibility, the staphylococcal strains were sub-cultured in BHI and grown to the logarithmic phase (0.6–0.7 OD_600_) at 37 °C, washed twice, and resuspended in phosphate buffered saline (PBS) (Sigma, Gillingham, UK). Cell suspensions (300 μL) were spread onto BHI agar and left to dry for 10 min. Antibiotic E-test strips (Biomerieux, Paris, France) were overlaid and plates were incubated at 37 °C for 24 h. Minimum inhibitory concentrations (MIC) (μg/mL) for each antibiotic were recorded in triplicate for each strain.

### 2.6. Antibiotic Treatment

Mice were weighed and anesthetized. Once sedated (as to not have the pedal withdrawal reflex), 20 µL of PAO1 was administered intranasally (i.n) (low dose 1 × 10^5^ or high dose 1 × 10^6^). The mice were left for 1 h and then tobramycin was administered by the Scireq InExpose nebulisation tower (Scireq, Montréal, QC, Canada). Mice were treated daily for 7 days in order to achieve complete bacterial clearance. Animals were sacrificed to assess bacterial burden at predetermined time points; lung, NALT, and fecal samples were collected.

### 2.7. Infection via the Drinking Water

Mice were administered 1 × 10^7^ CFU’s/mL PAO1 in their water bottle, the bacterial culture was changed every day for a fresh inoculum. After 5 days, the mice were given filtered water for the remainder of the study, prior to being sacrificed.

### 2.8. Cytokine Measurements

Cytokine levels were measured in lung homogenate supernatant using ELISAs as per manufacturer’s instructions. IL-6 and IL-1β were from R&D Systems (R&D, Minneapolis, MI, USA), IFNɣ and GM-CSF were from eBiosciene/Thermotech (eBioscience, Cheshire, UK; Thermotech, Nottingham, UK)

### 2.9. Flow Cytometry

Flow cytometry was performed as previously described [5,18]. Lung tissue was homogenised by passing through a cell sieve, the cells were obtained by spinning the homogenate at 600× *g* for 10 min at room temperature. The red blood cells were lysed with ammonium-chloride-potassium buffer and the cells were washed and counted using the Eve automated cell counter (NanoEntek, Waltham, MA, USA). The cells were incubated for 10 min in foetal calf serum (FCS) (100 µL) to block the Fc receptors. The cells were washed with 100 µL flow buffer (PBS, 10% FCS, 0.1% EDTA) centrifuged at 300× *g* for 10 min at room temperature, followed by extracellular and intracellular antibody staining. To identify neutrophils and macrophages, cells were stained with CD45 FITC, CD11b e660, CD11c e450, GR-1 PE and F4/80 PE Cy7 (eBiosience, Cheshire, UK). After incubation, the cells were washed twice and fixed with 4% Paraformaldehyde (PFA) Sigma Aldrich, Gillingham, UK). The fixed cells were stored at 4 °C and acquired within 48 h on a FACS Canto II flow cytometer (BD Biosciences, Wokingham, UK), analysis was performed with FlowJo software (Tree Star Inc., Ashland, OR, USA).

### 2.10. Assay for Alginate

Bacterial strains were grown in LB broth (Sigma Aldrich, Gillingham, UK), supplemented with 1 mM IPTG and carbenicillin (Sigma Aldrich, Gillingham, UK) for 48 h at 37 °C. Alginates present in lung supernatants were precipitated with cetyl pyridinium chloride (Sigma Aldrich, Gillingham, UK) and isopropanol (Sigma Aldrich, Gillingham, UK). Sample was subsequently mixed with 1 mL of borate-sulfuric acid (10 mM H_3_BO_3_ in concentrated H_2_SO_4_) (Sigma Aldrich, Gillingham, UK) and carbazole reagent (Sigma Aldrich, Gillingham, UK) (0.1% in ethanol) and incubated at 55 °C for 30 min and subsequently A500 was measured. As a standard, macrocystis pyrifera alginate (Sigma Aldrich, Gillingham, UK) was used.

### 2.11. Statistical Analysis

All CFU data was logarithmically transformed prior to analysis. GraphPad Prism (Version 9.4.1) was used to determine statistical significance between groups of mice. To compare two groups, an unpaired *t*-test was carried out, to compare three or more groups, an ANOVA test, with Sidak correlation for post-hoc multiple comparison applied. The duration of survival was calculated by Kaplan–Meier analysis, and statistical significance was determined with the log rank (Mantel–Cox) test. Error bars represent standard error of mean (SEM). The cut-off for statistical significance between groups was *p*
< 0.05.

## 3. Results

### 3.1. Clinical Isolate Chronic Infection Model

A major challenge in establishing chronic lung infection models is finding the balance between bacterial clearance and sepsis. Many models have relied on the use of artificial embedding materials to establish infection such as agar beads or slurry; however, in our hands these models are inconsistent, particularly in wild type C57 mice (Appendix A).

Several different strains of *P. aeruginosa* have previously been used to develop CF-like lung infections, with the laboratory strain PAO1, initially isolated from a wound infection, being the most commonly used strain [19]. However, PAO1 is more virulent than many isolates from patients with CF and has a faster replication [20]. Moreover, the motility of PAO1 in vivo has been reported to differ from that of clinical isolates from patients with CF [20]. Accordingly, we examined clinical isolates to determine their ability to establish chronic lung infection, without the necessity for embedding material. We have a biobank containing over 200 isolates obtained from the sputum of patients with and without CF, in addition to those isolated from patients within the intensive care unit (ICU). Following initial in vitro screening for; planktonic growth rate, biofilm formation and toxin production, a subset of these were selected for evaluation of their potential to chronically infect the murine lung.

Similar to previously reported studies [20], there was considerable variability in the virulence of the isolates, with three isolates significantly reducing survival rates of outbred CD1 mice, in comparison to infection with PAO1 infected mice (Figure 1A). In line with previously reported studies by Kukavica et al. [20], increased virulence largely corresponded with increased in vitro biofilm formation (Figure 1B), except for isolates Q502 and Q516. For Q516, this appears to be largely due low levels of alginate production in vivo (Figure 1C). Isolate Q502, obtained from the sputum of a patient with CF, appeared to be distinct from the other isolates, producing high levels of biofilm, both in vitro (Figure 1B) and in vivo (Figure 1C) but not causing significant mortality in the mice. This isolate was thus, selected for use to establish a chronic infection model.

In patients with CF, the acquisition of *P. aeruginosa* is often preceded by *S. aureus* infection [21,22,23]. We therefore sought to investigate if pre-infection with *S. aureus* influenced the susceptibility of the lungs to the development of chronic *P. aeruginosa* infection. As before, a range of clinical isolates obtained from the sputum of patients with CF, were utilised and compared to that of the commonly used Newman laboratory strain of *S. aureus*. Infection of CD1 mice with *S. aureus,* 10 days prior to infection with *P. aeruginosa,* resulted in a significant reduction in survival during PAO1 infection in comparison to mice given PBS prior to *P. aeruginosa* infection (Figure 2A), with the exception of isolate Q154 which enhanced survival (*p* = 0.04). Whilst the most significant effects on PAO1 infection were between pre-treatment with isolate Q154 and Q181, phenotypic characterisation of the isolates demonstrated differences in protease activity between strains (Appendix A).

Whilst many murine chronic infection models utilise C57BL6 or BALBc mice, it is well established that the murine strain selected for infection can have a significant impact on the outcome of a wide variety of infection models [24,25,26,27]. We therefore aimed to determine which mouse strain would be most permissive to the establishment of chronic infection (Figure 2B) with Q502. All strains rapidly succumbed to acute infection, the exception to this was Biozzi mice, in which 80% of mice were still alive at 144 h post-infection (Figure 2B). When Biozzi mice were pre-inoculated with *S. aureus* (Q154), 10 days prior to infection with *P. aeruginosa*, there was a significant reduction in *P. aeruginosa* bacterial burden in the lungs (*p* = 0.02) (Figure 2C), and reduced levels of the pro-inflammatory cytokine IL-6 (Figure 2D). Pre-treatment with *S. aureus* also resulted in significant increased levels of IFNγ and GM-CSF in the lungs, while IL-1β was unaffected (Appendix A). Moreover, there were no significant differences in the numbers of neutrophils and macrophages recruited to the lungs of mice pre-infected with *S. aureus,* compared to those that were not pre-infected (Appendix A).

At 10 days post-infection with *S. aureus* and prior to infection with *P. aeruginosa,* we confirmed that *S. aureus* was still present within the lungs (Figure 2E). Co-culture of *S. aureus* strains Q154 and Q502 with *P. aeruginosa* demonstrated delayed growth of *P. aeruginosa*, compared to that of when *P. aeruginosa* was cultured alone (Figure 2E) and statistical comparison of the AUC confirmed there was a significant impairment in growth (*p* = 0.002).

Pre-inoculation of Biozzi mice with Q154, prior to high dose *P. aeruginosa* infection, significantly enhanced their survival (*p* = 0.002) with 80% of the mice surviving to day 10 (Figure 2G). In Biozzi mice that received Q502 only, there was no detectable *P. aeruginosa* present in the lungs of mice surviving at day 14, suggesting they had cleared the infection, in contrast, in the animals pre-inoculated with Q154, there was a high burden of *P. aeruginosa* in 5 out 6 mice (average 3.4 × 10^3^ CFU ± 1.2 × 10^3^) at 14 days post-infection (Figure 2H).

### 3.2. Incomplete Antibiotic Clearance Leading to Chronic Lung Infection

In patients with CF, inhaled tobramycin is the standard treatment of chronic *P. aeruginosa* infection [28]. While generally effective in approximately 80% of patients, failed eradication is reported [29]. Utilising a clinically relevant treatment regimen of inhaled tobramycin, in our second model we aimed to develop a model of incomplete antibiotic clearance. Mice were infected i.n with 1 × 10^6^ CFU of PAO1 and subsequently treated with tobramycin for 5 days by nebulisation [30].

As expected, administration of nebulised tobramycin resulted in reduced bacterial burden in the lungs of mice 1 day post-treatment. Bacterial burden continued to decrease until day 5, and by day 7 mice had completely cleared the infection, as evident by the clearance of bacteria from the lung. However, by day 14, CFU counts returned to a level similar to that found at 1 day post-infection, and this was maintained up to 21 days post-infection (Figure 3A). Analysis of *P. aeruginosa* CFU in the lung and faeces of mice demonstrated that CFU was still detectable at 14 days post-infection and that there was a correlation between the lung CFU and the levels of bacteria being shed within the faeces (Figure 3B). This data demonstrates that a clinically relevant treatment regime of tobramycin nebulised therapy can be utilised in mice to induce chronic *P. aeruginosa* lung infection.

### 3.3. Water Bottle Chronic Infection Model

We next sought to investigate if the addition of PAO1 to drinking water could establish chronic lung infection. Animals exposed to the bacteria for 5 days, with a two day “wash-out” period to ensure that any *P. aeruginosa* detected was the result of an established infection, particularly in the upper respiratory tract, and not simply that the mouse that had recently taken a drink. On day 7, there was significant *P. aeruginosa* bacterial burden within the lungs of the mice. Following this, the CFU of *P. aeruginosa* detected within the lungs reduced until day 20, after which point it began to rise again (Figure 4A). In contrast, *P. aeruginosa* was detected in the NALT throughout all the time points tested (Figure 4B). Whilst levels of IL-6 in the lungs of mice were significantly increased across all time points tested, concurrent with the bacterial burden in the lungs, the highest levels of IL-6 were detectable at 7 days post-infection (Figure 4C).

As previously discussed, the current treatment regimen for treatment of *P. aeruginosa* infections with patients with CF is inhaled tobramycin [28]. However, *P. aeruginosa* infections often persist and reoccur. Thus, in our water bottle model, we administered mice tobramycin for 7 days by nebulisation to determine if the infection could be cleared. In this model, the mice remained clear following antibiotic treatment and, unlike treatment of high dose acutely infected mice (Figure 3), the treated mice still had no detected *P. aeruginosa* at day 50; this was not due to natural clearance as can be seen in the untreated mice at day 50 (Figure 4D). 

The βENaC mouse model accurately recapitulates many of the pulmonary features of CF and is commonly used to investigate CF pathogenesis [13,14,15,16]. It was demonstrated that the water bottle model could also be utilised to induce chronic *P. aeruginosa* infection in βENaC mice. As before, *P. aeruginosa* CFU was detected in the lungs of mice at day 7, which was maintained until day 50 (Figure 4E). Both βENaC and WT mice showed stable weight over the course of infection (Figure 4F), demonstrating that whilst the mice may have chronic infection and associated lung inflammation, they are not overtly symptomatic.

## 4. Discussion

The establishment of clinically relevant murine models of chronic lung infection has remained a challenge for many years and hampered our understanding of the pathogenesis of *P. aeruginosa* infection in patients with CF. Administration of *P. aeruginosa* to the murine lung either causes acute infection and subsequent sepsis or is rapidly cleared from the lung and, thus, cannot be used to establish chronic infection, as found in patients with CF [5]. Accordingly, alternative approaches have been developed for example, the embedment of *P. aeruginosa* within agar beads, or the coating of plastic tubes with *P. aeruginosa* for insertion into the bronchus [6,9]. However, such approaches are technically challenging, result in high mortality rates, and the agar bead model can induce varying inflammatory responses in the lung based on the beads used and their size [11]. In the present study, we describe for the first time three models of chronic lung infection which circumvent challenges associated with previously reported chronic models of *P. aeruginosa* chronic respiratory infection. These models allow infection to be sustained for longer time periods and importantly, the infections could be induced in both wild-type mice and in βENaC transgenic mice [31].

By careful selection of clinical isolates of *S. aureus* and *P. aeruginosa*, we were able to establish chronic infection in Biozzi mice. Interestingly, a correlation between virulence and high biofilm formation was evident across the tested strains, with the exception of Q516. This isolate, however, demonstrated a significant reduction in the levels of alginate detected in vivo, compared to that of PAO1, demonstrating that in vitro biofilm assays may not fully recapitulate the in vivo conditions. This is in line with previously reported studies [32], [33]. Jordan and co-workers reported nonconformity of biofilm formation with several strains of *S. aureus* in vitro and in vivo. Specifically, it was reported that deletion of the accessory gene regulator (*agr*) resulted in increased biofilm in vitro but did not induce increased biofilm in vivo or affect mortality rates in mice [33]. To date, there have been no reported studies comparing biofilm formation of isolates of *P. aeruginosa* either in vitro or in vivo. A number of studies have reported that murine strain can significantly impact the severity of infection against a number of bacteria [24,25,26,27]. In this model of PAO1 infection, Biozzi mice were most resistant, with 80% surviving at 144 h post-infection, while all other strains rapidly succumbed.

In patients with CF, both *S. aureus* and *P. aeruginosa* are key bacterial pathogens. While chronic infection with *P. aeruginosa* is accepted to be the leading cause of airway infection and is associated with poorer outcomes, the impact of *S. aureus* colonisation on subsequent outcome to *P. aeruginosa* remains debated [34]. Cigana eat al. utilised the agar bead model to infect mice with *S. aureus*, prior to infection with *P. aeruginosa* [34]. However, as previously described, the agar bead model has many disadvantages. We therefore sought to determine if pre-exposure to *S. aureus* would render the lungs more permissive to chronic infection by *P. aeruginosa.* Whilst there was significant variability in the impact of *S. aureus* isolates on the outcome of the subsequent *P. aeruginosa* infection, we were able to identify an isolate, Q154, which enhanced survival of the mice and resulted in the establishment of chronic infection. *S. aureus* was still maintained in the lungs at 10 days post-infection, prior to infection with *P. aeruginosa,* and this prompted us to determine if there was a direct interaction between the bacterial species. Interestingly, in vitro culture of *S. aureus* strains with *P. aeruginosa* demonstrated impaired growth of *P. aeruginosa*. This is in line with previously reported studies by Michelson et al., who demonstrated that *S. aureus* altered the growth activity and autolysis of a range of *P. aeruginosa* isolated in vitro [35]. Further studies are warranted to fully investigate the interspecies interactions between isolates of *S. aureus* and *P. aeruginosa* obtained from patients with and without CF and to fully understand host adaptation in the CF lung.

Whilst the model outlined above is highly biologically relevant and recapitulates many of the features of establishment of chronic lung infection seen in patients, it has its limitations. Firstly, it requires the use of specific clinical isolates; Q502 is highly antibiotic resistant and therefore genetic manipulation of the bacteria to either make it bioluminescent to monitor infection in real time or create specific mutants to investigate the virulence factor is technically demanding. Furthermore, it requires Biozzi mice; the model does not readily transfer to C57BL6 models. Although Biozzi mice are commercially available, this is not a widely utilised strain of mice and transgenic models are not available on this genetic background, increasing the time and costs required to use transgenics to investigate host-pathogen interactions.

The antibiotic treatment model was discovered serendipitously; however, it represents a biologically relevant model. The model uses C57BL6 mice making it applicable to study how immunological manipulation of the host will impact the establishment of chronic infection. Whilst here we have used PAO1, clinical isolates can also be used and their clearance, bounce back, and ability to establish chronic infection can be easily investigated. It has recently been reported that antibiotic exposure induces both phenotypic and genotypic alterations in *P. aeruginosa* [36]; this model would make an excellent tool to further examine this phenomenon. It is interesting to note that mice treated with antibiotics following either low dose infection (Appendix A) or infection via the water bottle either do not show any bounce back or else do have sustained levels of bacteria within their lungs. This would suggest that there is a threshold level of bacteria required for chronic infection to be maintained. Singh et al. reported that there is a bacterial threshold within the lung for induction of inflammation [37]; perhaps the inflammatory response contributes to the establishment of infection, mirroring the role that inflammation plays in driving microbial dysbiosis and the establishment of chronic infection within the gut. We demonstrated that there is a correlation between *P. aeruginosa* CFU within the lung and bacterial fecal shedding; this observation provides a useful means for monitoring chronically infected mice.

It has previously been reported that the water bottle method of induction of chronic lung infection is only suitable for transgenic murine models of CF [19]. However, in our adapted protocol this is not the case, and we describe the successful establishment of chronic infection in wild-type mice. This model facilitates the investigation of spontaneous infection and the pathogen virulence or host factors that modulate the initial infection or the development of chronic infection. We also demonstrate successful induction of chronic infection in the βENaC mouse model of CF, thus facilitating investigations into the role of chronic infection in CF pathogenesis. Whilst *P. aeruginosa* is identified as a pathogen of patients with CF, it is associated with other respiratory diseases including non-CF bronchiectasis and chronic obstructive pulmonary disease [38,39,40]. Moreover, it has been reported that persistent *P. aeruginosa* infection is a risk factor for chronic rejection in allograft lungs [41]. Therefore, our model in wild-type mice, rather than genetically modified mice, could also be utilised for the study of non-CF related chronic infections.

## 5. Conclusions

*P. aeruginosa* remains an important cause of chronic respiratory infections in CF and non-CF lung diseases. Its ability to evade host responses results in significant morbidity and mortality. The murine models described herein present reliable and useful models to study chronic *P. aeruginosa.* The models present all recapitulate different aspects of clinical disease, and, when combined, they provide an essential tool set to investigate the role of both pathogen and host factors that can modulate the establishment or persistence of chronic infection. These tools are essential to increase our understanding of the biological processes involved, as well as identify novel therapeutic targets to tackle chronic infection.

## Figures and Tables

**Figure 1 pathogens-12-01053-f001:**
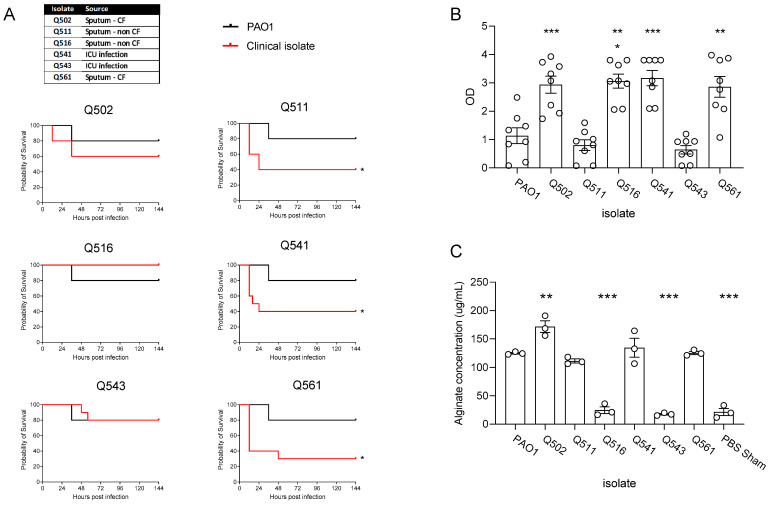
CD1 Mice were infected intranasally with clinical isolates of *P. aeruginosa* and compared to that of the common laboratory strain, PAO1 (1 × 10^7^). (**A**) Survival was monitored (n = 10 per strain). (**B**) In vitro biofilm production of the clinical isolates and PAO1 was measured in vitro (n = 8 replicates per strain) and (**C**) in vivo alginate production was measured. (n = 8 per group). ◯ represent individual data points, * *p* ≤ 0.05, ** *p* ≤ 0.01, *** *p* ≤ 0.001.

**Figure 2 pathogens-12-01053-f002:**
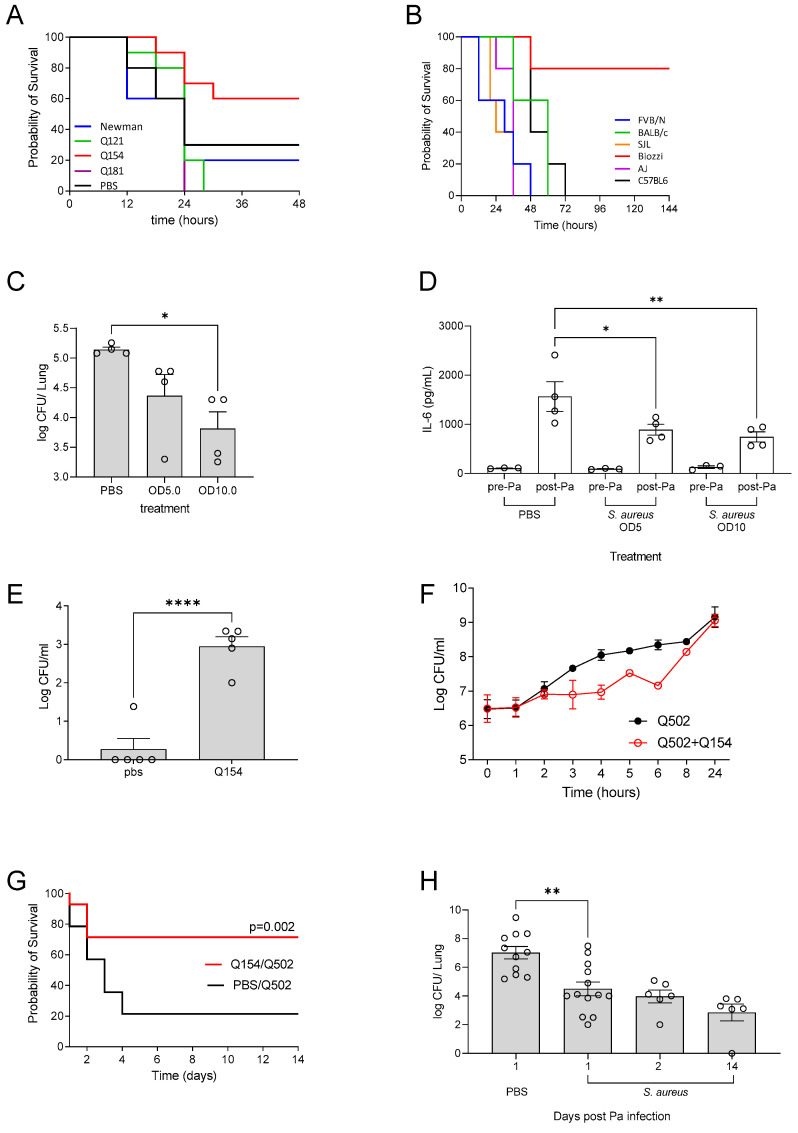
(**A**) Pre-treatment (at -10 days) of CD1 mice with *S. aureus* (OD5) prior to *P. aeruginosa* infection—clinical isolates of *S. aureus* all from sputum samples from patients with CF, control animals were pre-treated with PBS. (**B**) Comparison of inbred mouse strains infected with *P. aeruginosa* Q502 1 × 10^7^. (**C**) The bacterial CFU following pre-treatment of Biozzi mice with *S. aureus* Q154 OD5 and OD10 or control PBS prior to *P. aeruginosa* infection and (**D**) IL-6 levels pre and post *P. aeruginosa* infection. (**E**) Detection of Q154 within the lungs 10 days after inoculation. (**F**) In vitro co-culture of Q502 and Q154. (**G**) Infection of biozzi mice with high dose (1 × 10^7^) Q502—prior inoculation with Q154 significantly enhances survival and (**H**) there was detectable *P. aeruginosa* in the lungs 14 days post-infection. ◯ represent individual data points, * *p* ≤ 0.05, ** *p* ≤ 0.01, **** *p* ≤ 0.0001.

**Figure 3 pathogens-12-01053-f003:**
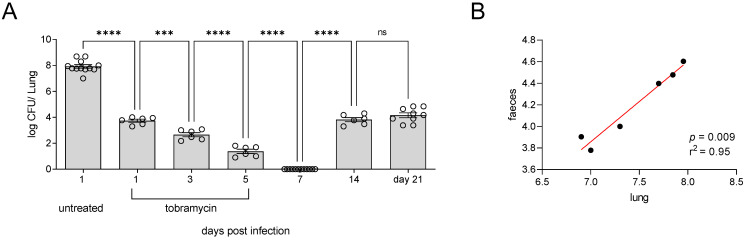
(**A**) Mice were infected intra-nasally with 1 × 10^6^ CFU of *P. aeruginosa* (PAO1). At 1 h post-infection, mice were administered inhaled tobramycin via nebulisation (10 mg/kg) for 7 days. At each time point, mice were sacrificed for enumeration of CFU in the lungs. (**B**) At 14 days post-infection, lung and faeces were collected; there was a significant correlation between bacterial levels in the lung and the amount being shed in faeces. Untreated n = 12, treated n = 6–12 per time point, *** *p* ≤ 0.001, **** *p* ≤ 0.0001.

**Figure 4 pathogens-12-01053-f004:**
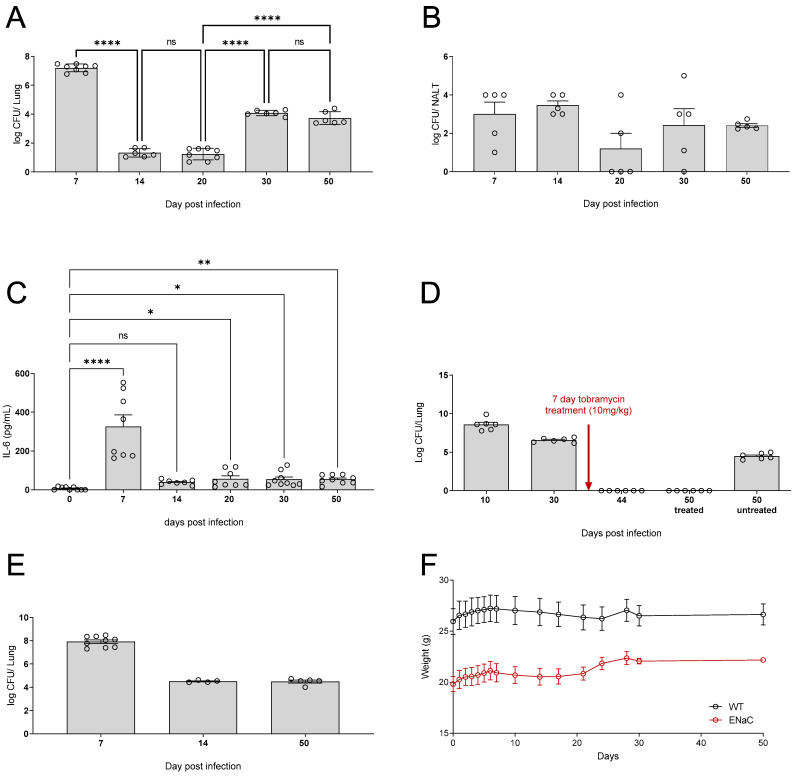
C57BL6 mice infected with PAO1 in water bottle (1 × 10^7^ per ml) for 5 days. Mice were then transferred back onto fresh drinking water and culled at days 7, 14, 20, 30, and 50 and the levels of CFU in (**A**) the lungs (**B**) and NALT. (**C**) The levels of IL-6 in the lungs were quantified (n = 6–9 per time point). (**D**) Following treatment in this model, there is sustained bacterial clearance in water bottle infected mice (n = 6 per time point). βENaC mice were infected with PAO1, and the CFU in (**E**) the lungs was quantified and the (**F**) weights were measured (n = 6 per time point). ◯ represent individual data points, * *p* ≤ 0.05, ** *p* ≤ 0.01, **** *p* ≤ 0.0001.

## Data Availability

Not applicable.

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
