# Peer review of "Biologically Relevant Murine Models of Chronic Pseudomonas aeruginosa Respiratory Infection"

_pathogens, 2023, doi:10.3390/pathogens12081053_

Round 1

Reviewer 2 Report

In this manuscript Rogers et al., describe three models of chronic lung infection with P. aeruginosa. Overall, the study constitutes a significant methodological advance, is well documented, and the authors discuss the advantages as well as limitations of their models. I must say that I enjoyed reading it and I only have a few minor remarks.

1. line 109 what medium was used for outgrowth?. Was there a normalization e.g. per gram of lungs?. If so, please indicate the normalization  on y-axes on figures (e.g. 2C)

2. Biofilm data (Fig 1B) - please clarify weather this readings (OD 595) were normalized to OD 600 nm (i.e. culture density). Were there differences in the growth of the strains under these conditions?  

3. In figure legends, for clarity of the presentation it might be good to add a summarizing  figure titles  and move letters e.g. (A) to the beginning of sentences.

4. Line 198 - Q511 - should be Q516?, line 211 delete 'were', line 283 missing p=,  line 176 - 'sidak's'

5. 217 - 'no significant differences' please rephrase - there is a difference between the strains( in protease activity (Table S1)) but its just the opposite as one would expect.

6. Alginate quantification protocol (Fig. 1C) is not described in materials and methods?

7. e.g. line 267, 284, or 301 consider changing 'CFU' to P. aeruginosa CFU or just P. aeruginosa

Reviewer 3 Report

General comment

The work has been done well and the below minor issues need to be rectified.

 Parts that needs revision are:

 2. Materials and Methods

            2.5. Antibiotic treatment;

L 146-147: Needs explanation; Is one hour enough for bacteria to establish itself in the lungs? Is it possible that the drug could have cleared the bacteria before it establishes itself appropriately in the lungs?

L147: Please provide details of this equipment's (…scireqInExpose nebulisation tower…) source, i.e. either indicate the company of origin, city and Country or state the products catalogue number and company name.

 2.7. Cytokine measurements

L156-7: “... were from R&D Systems…” and “…were from eBiosciene/Thermotec.” This is not enough. Please provide more details of this reagents' source, i.e. either indicate the company of origin, city and Country or state the products catalogue number and company name.

2.8. Flow cytometry

L168: Provide details of these staining reagents' source, i.e. either indicate the company of origin, city and Country or state the products catalogue number and company name.

 3. Results

           3.2. Incomplete antibiotic clearance leading to chronic lung infection

            L255-6, 269: “P. aeruginosa” needs to be italicized.

3.3. Water-bottle chronic infection model

            L280: grammatical error (delete “…were…”)

 4. Discussion

L339: ... impact "he"... is a typographical error

 .
